# MPCI: A novel metric for quantifying DNA methylation patterns in NGS data

**Naghme Nazer**[1], **Hoda Mohammadzade**[1], **Mahya Mehrmohamadi**[2]*

1 Department of Electrical Engineering, Sharif University of Technology, Tehran, Iran, 2 Department of Molecular Biotechnology, School of Biotechnology, College of Science, University of Tehran, Tehran, Iran

* mehrmohamadi@ut.ac.ir

## Abstract

Epigenetic processes, particularly disruptions in DNA methylation profiles, are associated with many disease states. Traditional approaches for DNA methylation biomarker discovery focusing on individual CpG sites do not account for fragment-level methylation states. Methylation haplotype analysis offers a more comprehensive approach leading to increased distinction capability between reads originating from tissues with diverse methylation profiles. This can be particularly valuable in liquid biopsy where detecting small amounts of disease-specific cell-free DNA (cfDNA) amidst a bulk of healthy cfDNA is challenging. To address limitations of existing metrics for quantifying methylation patterns in a region from sequencing data, we propose the Methylation Pattern Consistency Index (MPCI), a novel metric that captures consistent methylation patterns across sequencing reads, accounting for both methylated and unmethylated blocks of CpGs. Using whole-genome bisulfite sequencing data, we demonstrate that MPCI outperforms MHL and its symmetric counterpart, dMHL (MHL – uMHL), across several benchmarks: distinguishing closely related cell types (CD4 vs. CD8; AUC 0.915), multi-tissue classification (0.92 accuracy), and detection of in-silico cfDNA spike-ins at abundances as low as 1%. Notably, in a clinical liquid-biopsy cohort of liver transplant patients, MPCI achieved significantly higher classification performance than dMHL (Accuracy: MPCI: $0.868 \pm 0.023$ vs. dMHL: $0.768 \pm 0.027$, $p = 0.014$) in discriminating pre- from post-transplant cfDNA profiles. These findings position MPCI as a reliable quantification approach for biomarker selection or diagnostic testing in epigenetic studies. We have made MPCI available as an R function for usage convenience.

## Author summary

DNA methylation is a chemical tag added to DNA that helps control how genes are turned on and off. Disruptions to these patterns are common in diseases such as cancer, and traces of these changes can be detected in the small

**Data availability statement:** All datasets used in this study are publicly available from the GEO database under the accession codes referenced in the manuscript (GSE186458, GSE262275, GSE43414). All custom R scripts used for data analysis, simulation, metric calculation (including the MPCI function), and figure generation are publicly available under an open-source license at: https://github.com/NaghmeNazer/MPCI.

**Funding:** This work was supported by the Iran National Science Foundation (grant number 99012262 to MM), and Research and Technology Office of Sharif University of Technology (SUT) (to HM and NN). The funders had no role in study design, data collection and analysis, decision to publish, or preparation of the manuscript.

**Competing interests:** The authors have declared that no competing interests exist.

fragments of DNA circulating in blood—a technique known as liquid biopsy. A major challenge, however, is that disease-derived signals are often overwhelmed by abundant DNA from healthy cells.

We developed a new computational approach, the **Methylation Pattern Consistency Index (MPCI)**, which identifies consistent methylation patterns—both present and absent—across neighboring DNA sites in sequencing data. We show that MPCI outperforms existing methods by more effectively distinguishing closely related cell types, detecting tissue-derived DNA at levels as low as 1%, and robustly tracking changes in real patient liquid biopsy samples.

MPCI provides researchers and clinicians with a more sensitive and balanced way to quantify epigenetic signals. This approach has the potential to improve early disease detection, enhance treatment monitoring, and increase the accuracy of liquid biopsy–based diagnostics across a wide range of conditions.

## Introduction

Epigenetic processes play crucial roles in gene regulation. One of the most extensively studied epigenetic mechanisms is DNA methylation, where a methyl group is added to the 5-carbon of cytosine residues, typically within CpG dinucleotides. This modification can influence gene expression by recruiting proteins that are involved in transcription regulation or by changing chromatin compaction and affecting protein bindings [1]. The methylation status of specific regions is tissue-specific and dynamic, and its abnormal alteration is associated with many disease conditions [2–4]. The relative stability of DNA methylation marks as well as the ease of assaying them experimentally make DNA methylation alterations invaluable biomarkers for various clinical applications [5–7].

In the context of disease-associated DNA methylation alterations, selecting CpG sites or regions with differential methylation is the main step of biomarker discovery. Due to the inherent noise in individual CpG site analysis, and given the fact that methylation states of nearby CpGs in the genome are highly correlated [8], marker design increasingly benefits from considering methylation haplotypes - the methylation pattern of a set of adjacent CpGs [9]. Capturing methylation signatures across a set of correlated CpGs provides a more robust and consistent epigenetic signal leading to improved limit of detection and increased accuracy [10].

An important clinical context where analyzing methylation haplotypes instead of single CpGs proves highly advantageous is liquid biopsy. Liquid biopsy refers to the minimally-invasive analysis of cell-free DNA (cfDNA) fragments circulating in the bloodstream for detection of abnormal conditions in the body, that has proven valuable for early disease detection, prognosis, and treatment monitoring [11,12]. In these assays, the key challenge often lies in detecting a small fraction of tissue-specific or disease-associated cfDNA amidst a large background of cfDNA originating from

healthy tissues [13]. Individual CpG sites are often hard to distinguish from noise, especially in the low-abundance cfDNA context. In contrast, haplotype-based analysis captures the consistency of methylation across multiple CpG sites from a sequencing read, increasing the likelihood of detecting true tissue-specific signals even at low concentrations [14,15]. This approach becomes particularly important when analyzing liquid biopsy samples with low tumor content, as is the case in early cancer detection or minimal residual disease monitoring.

A primary step in extracting DNA methylation information from next generation sequencing (NGS) read data is region-wise quantification of methylation. Several metrics are used in the field to convert methylation states of CpGs in a genomic region into numerical values for downstream analyses [15–20]. Methylation Frequency (beta-value), one of the most commonly used metrics for calculating methylation in a CpG, measures the proportion of methylated CpGs out of the total CpGs present from the pile-up of all sequencing reads that cover a given site. While straightforward and easy to understand, methylation frequency only accounts for methylation ratio and does not consider the pattern of methylation over adjacent CpG sites. When quantifying methylation in a region, the average of individual CpG methylation ratio is typically used which has limited sensitivity in capturing small changes in a sample, e.g., low tumor fraction in cfDNA. Therefore, researchers have proposed alternative metrics to address these limitations.

Among alternative metrics to the methylation frequency, methylation entropy (ME) [16], epi-polymorphism (EP) [17], proportion of discordant reads (PDR) [18], fraction of discordant reads pairs (FDRP) and quantitative FDRP (qFDRP) [19] describe heterogeneity of methylation patterns between reads. Methylation concordance (MeConcord) introduced in a recent study basically consists of two separate metrics, Reads Concordance (RC) and CpG Concordance (CC), that are collectively used to quantify local DNA methylation concordance using Hamming distance [20]. While effective in distinguishing methylation patterns, its utility has not been investigated in low fraction signal contexts such as cfDNA in liquid biopsy applications [20]. Methylation Haplotype Load (MHL) is defined as weighted mean of the fraction of fully methylated substrings at different lengths for quantification of the co-methylation of adjacent CpGs in a region. MHL has demonstrated superior performance in addressing the challenge of capturing methylation patterns across regions with similar methylation frequencies but different patterns [15], making it a well-established and widely used metric in epigenetic studies. Its reliability has been validated in other studies [21,22] and has been used in different applications of methylation analysis [23,24].

Nevertheless, MHL has certain drawbacks that motivate the development of yet additional metrics. [25,26]. Firstly, MHL evaluates the consistency of consecutive CpG sites within a haplotype but fails to account for the consistency of these haplotypes across different reads at the same locus. Secondly, MHL focuses solely on methylated CpG patterns and disregards unmethylated CpG sites, necessitating additional measurement of unmethylated haplotype load (uMHL) for comprehensive methylation quantification [22,24]. Relying solely on MHL or uMHL may result in inconsistent outcomes when distinguishing between hypo-methylated and hyper-methylated markers. In other words, in a completely symmetric scenario, MHL might favor different regions depending on whether the control region is methylated or unmethylated. To address these issues, we introduce a single, symmetric metric capable of effectively capturing diverse scenarios. Here, we propose the Methylation Pattern Consistency Index (MPCI) and show that our novel metric has improved ability over MHL and its symmetric counterpart, dMHL (MHL − uMHL), in capturing diverse methylation signals such as rare methylation haplotype detection in NGS data.

## Results

### Definition of MPCI (Methylation Pattern Consistency Index)

We propose methylation pattern consistency index (MPCI) as a symmetric measure for quantifying methylation patterns across sequencing reads and consecutive CpG sites in methylation haplotypes by computing weighted manhattan similarities (Fig 1a, See Methods). Fig 1b and 1c demonstrate an example scenario where two cases with symmetric patterns of CpG methylation exhibit distinct behaviors based on MHL and uMHL. When assessing a hypo-methylated region, MHL

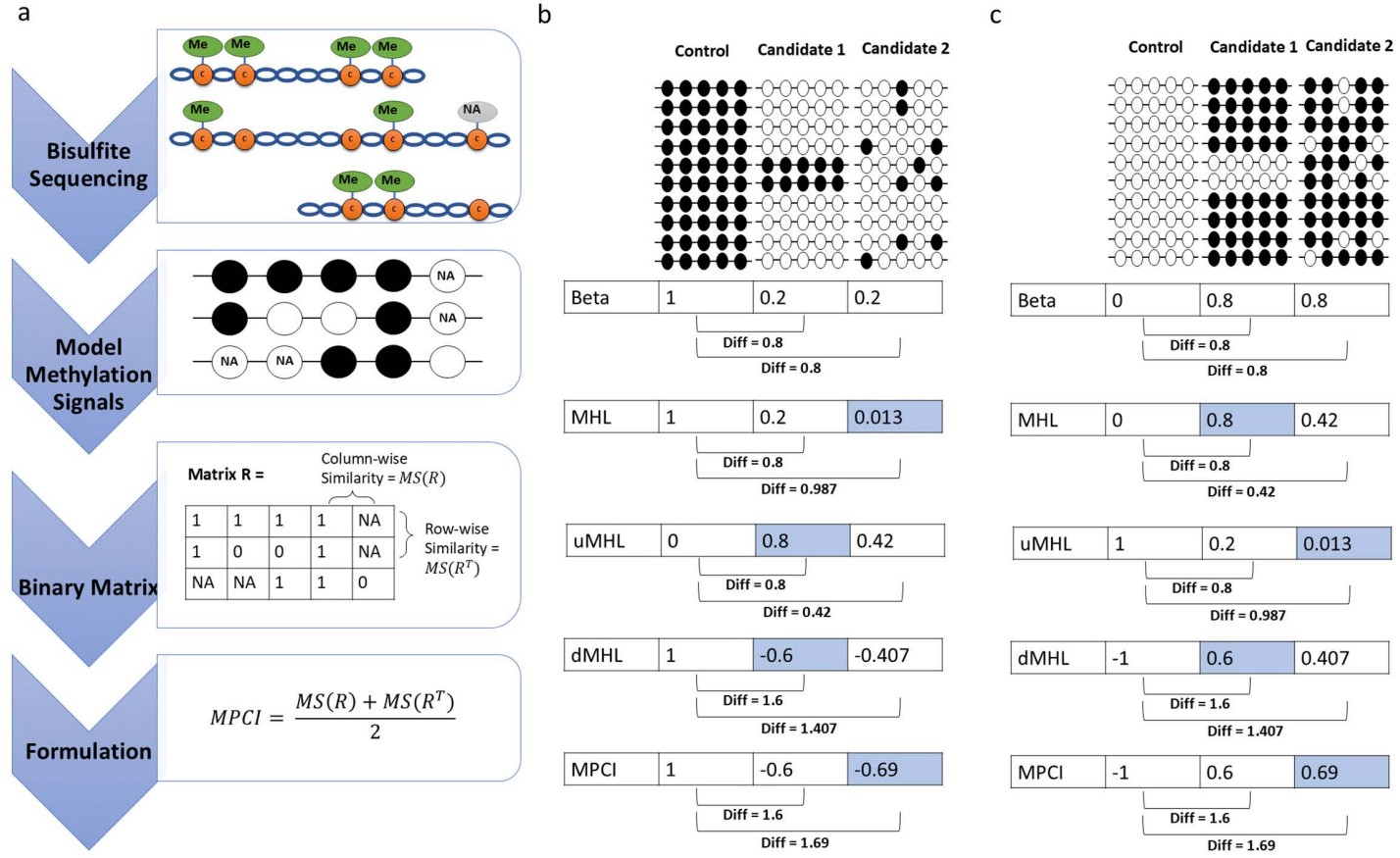

**Fig 1. a) Workflow for converting NGS-derived methylation data to MPCI. Schematic representation of the binary matrix R where rows correspond to sequencing reads and columns correspond to consecutive CpG sites.** Methylated CpGs are represented as 1s (black) and unmethylated CpGs as 0s (white). Unknown methylation status is represented with NA values. MS stands for Manhattan Similarity (See Methods). b) Comparison of symmetry and consistency across metrics in hypo-methylated marker selection. There are two candidate regions contrasted against the fully methylated control region. Blue cells show where MHL/uMHL behave inconsistently. c) Comparison of symmetry and consistency across metrics in hyper-methylated marker selection. There are two candidate regions contrasted against the fully unmethylated control region. Blue cells show where MHL/uMHL behave inconsistently.

reveals a significant difference between a region with randomly methylated CpGs and a fully methylated control region (Fig 1b). However, its behavior diverges entirely when hypermethylated markers are considered (Fig 1c). When uMHL is employed for marker selection, it shows a strong difference between a region with randomly unmethylated CpGs and a fully unmethylated control region in hyper-methylation discovery settings (Fig 1c). For hypo-methylated marker selection, uMHL favors regions with strong methylated haplotypes (Fig 1b). This analysis highlights that when using MHL and uMHL for marker selection, prior knowledge of methylation differences between the two groups is essential. However, such information is typically unavailable before marker discovery analysis and may necessitate additional pre and post selection analysis and careful interpretation. In these examples, MPCI and differential MHL (dMHL = MHL – uMHL) behave similarly (Fig 1b and 1c), showing both these metrics are able to capture similar aspects of variation.

## Performance of MPCI in cell-type classification contexts

Next we set out to assess MPCI as a metric for real-life contexts of biomarker discovery. Publicly available Whole Genome Bisulfite Sequencing (WGBS) data from a variety of healthy human tissues and cell types were obtained from

GEO [27]. To ensure unbiased analysis, 400 random genomic regions with a diverse range of total CpG numbers (3–22 CpGs) were chosen and 392 non-overlapping regions were retained for downstream analyses. Two closely related cell types (CD4 and CD8 T lymphocytes) with small methylation difference were selected. The goal of this experiment was to determine which metric better captures the subtle methylation differences between these cell types.

The performance of MPCI was benchmarked against seven previously established metrics: Methylation Haplotype Load (MHL), Unmethylation Haplotype Load (uMHL), Methylation Entropy (ME), Epi-polymorphism (EP), Proportion of Discordant Reads (PDR), Fraction of Discordant Read Pairs (FDRP), and Quantitative FDRP (qFDRP) [19]. Upon calculating these metrics for the selected regions across samples, eight distinct datasets were generated for classification. Principal Component Analysis (PCA) demonstrated that MPCI achieves competitive performance in separating CD4 and CD8 samples compared to the other metrics (S1 Fig). Subsequently, binary classification (CD4 vs CD8) was performed using a Support Vector Machine (SVM) with nested cross-validation (5 outer folds, 3 inner folds) to optimize hyperparameters and prevent data leakage. Model performance was evaluated using AUC, accuracy, specificity, and sensitivity, with statistical significance between metric pairs determined via non-parametric Wilcoxon rank-sum tests (Fig 2a). MPCI achieved the highest mean AUC ($0.915 \pm 0.15$) (Fig 2b). MPCI showed statistically significant improvement in comparison with all metrics except qFDRP and uMHL. Across all other performance metrics (Accuracy, Sensitivity, Specificity), MPCI consistently ranked as the top performer. Specifically, MPCI achieved the highest Accuracy (0.740), Sensitivity (0.720), and Specificity (0.760) when only these 392 random regions were used (S1 Table). These results indicate that MPCI is more effective at capturing methylation differences between closely related cell types, making it a superior metric for biomarker discovery in such epigenetic studies.

Next, to test how sensitive our performance comparisons are to region selection, we repeated the CD4 vs. CD8 classification analysis using Methylation Haplotype Block (MHB) regions as previously introduced [15] and compared the results to our original analysis using randomly selected regions. Both MPCI and MHL performance improved within MHBs, consistent with the known co-methylation structure of these regions. However, MPCI still significantly outperformed MHL in AUC ($p = 0.027$) and sensitivity ($p = 0.006$) even within this MHL-favored context (S2 Table). The performance advantage of MPCI over MHL was greater in the unbiased, random region context, highlighting MPCI's superior generalizability for applications without prior knowledge of optimal genomic regions.

To further evaluate MPCI in a broader classification context, a multi-class classification task was performed using samples from brain, liver, colon, lung, and bone marrow tissues from the same study [27]. These tissues were selected for their diverse methylation patterns and relevance to disease contexts. MPCI and MHL were calculated for the same set of 392 random regions in these samples. A multi-class SVM classifier was trained on 80% of the samples, and accuracy was evaluated on the remaining 20%. This procedure was repeated 100 times. MPCI achieved an average accuracy of 92.25% ($\pm 0.88\%$), outperforming MHL, which reached an average accuracy of 87.62% ($\pm 1.1\%$) (S2 Fig). These findings further demonstrate MPCI's robustness and versatility in distinguishing methylation patterns across diverse tissue types, highlighting its potential for applications in tissue-specific biomarker discovery and disease detection.

### Definition of Differential MHL (dMHL)

MPCI and MHL are intrinsically different in that MPCI is symmetric with respect to methylation states, but MHL is defined as the level of consecutive co-methylation. To further make these two metrics more directly and fairly comparable for downstream evaluations, we defined a differential MHL metric as $dMHL = MHL - uMHL$. This formulation can be viewed as a simple symmetric extension of the original MHL. The relationship between MPCI and dMHL was then assessed in CD4 and CD8 cell types. Our analysis revealed a strong positive correlation between MPCI and dMHL ($R^2 = 0.81$, $p < 0.001$), indicating a substantial statistical association between the two metrics. Nevertheless, the presence of considerable unexplained variance and systematic deviations suggests that MPCI captures information not fully represented by dMHL, and vice versa (S3 Fig).

PLOS Computational Biology

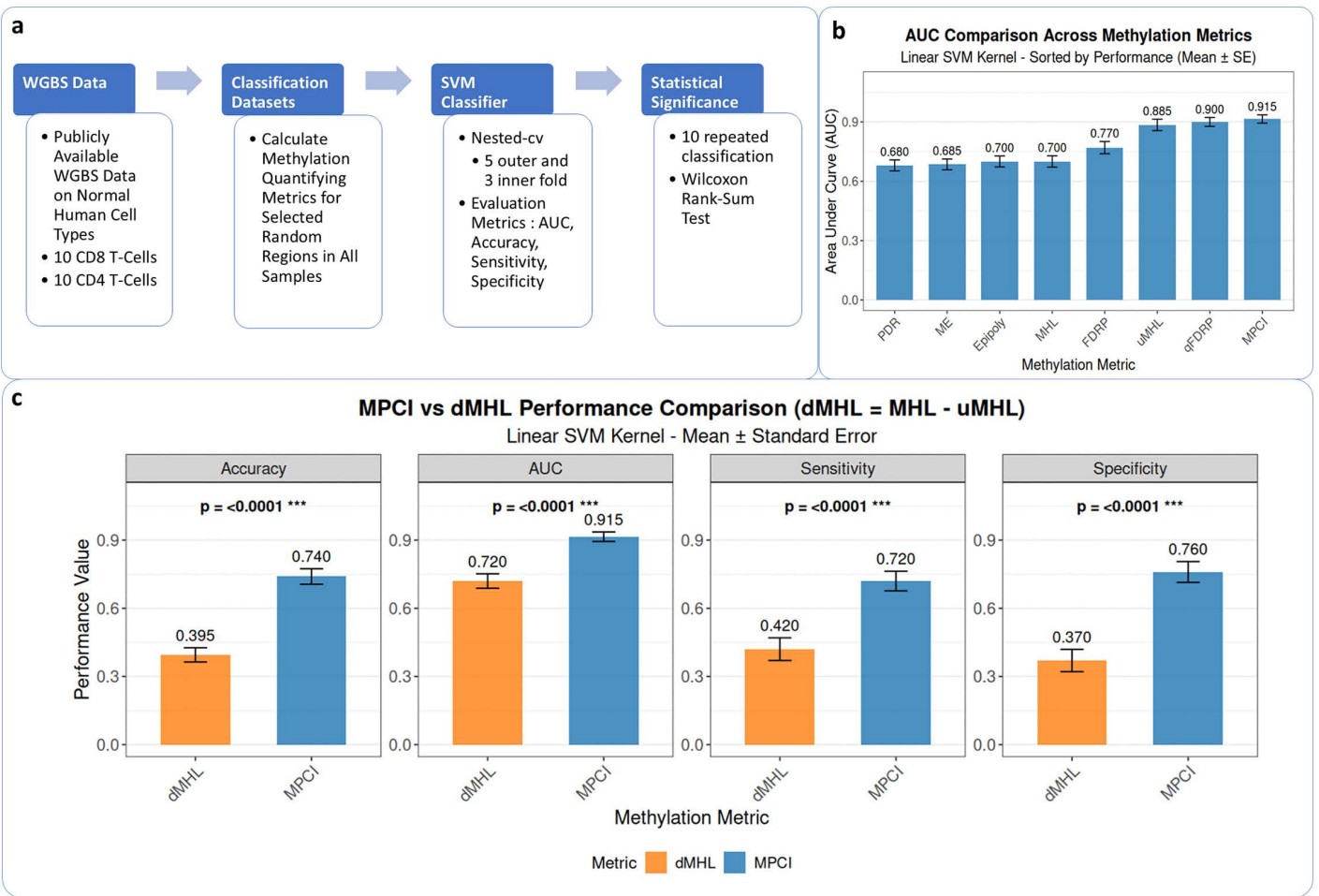

**Fig 2. a) Workflow of performance evaluation in cell type classification across metrics.** Public methylation sequencing data from healthy human cell types was used, focusing on two closely related cell types (CD4 and CD8). Ten samples of each were selected, and eight methylation quantifying metrics were calculated for random regions. SVM classifier performance metrics (AUC, accuracy, specificity, sensitivity) were calculated. This process was repeated 10 times and statistical significance of all comparisons was evaluated using Wilcoxon rank-sum test. b) AUC comparison across methylation metrics. The bar plot represents the mean AUC of the classifier. Error bars represent the standard error. c) Comparison of MPCI and dMHL in CD4 vs. CD8 methylation analysis. Bar plots comparing the performance of MPCI (blue) and dMHL (orange) in distinguishing CD4 vs. CD8 cells. Error bars represent standard errors.

Based on these observations, our comparative analyses focused on MPCI versus dMHL, as dMHL aligns better with MPCI and simultaneously incorporates both methylated and unmethylated blocks. We compared the performance of MPCI and dMHL in the same CD4 vs. CD8 classification task. The results demonstrate that MPCI significantly outperforms dMHL across all performance metrics. The performance gap is particularly pronounced in Accuracy and Specificity, where MPCI shows improvements of 0.345 and 0.390, respectively. These differences are highly statistically significant (all p < 0.001) (Fig 2c).

## Performance of MPCI in disease detection contexts

Next, to evaluate the performance of MPCI in detecting lowly abundant disease-specific methylation patterns in NGS data, we analyzed plasma cell-free DNA methylation [27]. We used data from healthy cfDNA and healthy brain tissues,

and simulated cfDNA in neurodegenerative diseases, where presumably a fraction of plasma cfDNA originates from brain tissue and is present at low concentrations in the bloodstream [28]. For this experiment, 10 brain-related samples and 23 healthy cfDNA samples were obtained [27]. In-silico spike-in of brain-originating reads into healthy cfDNA reads were simulated in 1–10% fractions (See Methods). To ensure an unbiased comparison, 392 randomly selected and non-overlapping genomic regions were used. Healthy samples were generated by randomly sampling reads from the 23 healthy cfDNA dataset for these regions. For the disease samples, reads from 14 brain tissue samples were randomly selected based on the desired spike-in ratio (ranging from 1% to 10%) and substituted into the healthy cfDNA data. This process resulted in 100 healthy and 100 disease samples and was repeated 5 times to check for reproducibility (Fig 3a). MPCI, MHL and dMHL were calculated for all regions in these samples, and SVM classifiers were built to distinguish between the two classes. This process was repeated for all 5 sets of simulations and the results of the classifiers were aggregated (see Methods).

Results demonstrate that integrated/symmetric methylation metrics (MPCI and dMHL) significantly outperform MHL alone across all spike-in ratios. MPCI consistently achieved superior mean performance compared to both dMHL and MHL, with the performance advantage becoming more statistically robust at higher spike-in concentrations. At a 1% spike-in ratio, MPCI attained an AUC of 0.567 ($\pm$0.063), outperforming dMHL (0.564$\pm$0.058) and MHL (0.557$\pm$0.058). As spike-in ratios increased (3–10%), MPCI not only maintained superior mean performance but also demonstrated substantially lower standard deviations than competing metrics, indicating enhanced measurement stability. At 10% spike-in, MPCI achieved near-perfect discrimination (AUC: 0.9995$\pm$0.0016) with significantly less variability than dMHL (0.9952$\pm$0.0094) and MHL (0.9908$\pm$0.0127) (Fig 3b). These results highlight MPCI's dual advantage: superior discriminatory power and increased statistical robustness. These combined position MPCI as an optimal metric for detecting disease-specific cfDNA across a range of clinically relevant concentrations. Results were similar for other performance metrics of accuracy, sensitivity, and specificity (S3 Table).

## Evaluation of MPCI on external liquid biopsy data

To validate MPCI beyond simulations and in real clinical samples, we evaluated its ability to distinguish pre- and post-operative cfDNA methylation profiles in liver transplant patients. Using publicly available WGBS data (GSE262275) [29], we analyzed 58 samples collected before (PRE, n=27) and immediately after transplantation (PRE, n=31). High-coverage genomic regions were selected, and MPCI was computed alongside dMHL for comparison (see Methods). A nested cross-validated SVM classifier was trained to discriminate between pre- and post-transplant states. Statistical comparisons between MPCI and dMHL were performed using Wilcoxon rank-sum tests on cross-validation results, with significance defined as $p < 0.05$ after Benjamini–Hochberg adjustment. Mean AUC values were calculated from standard errors across folds (See Methods).

MPCI demonstrated superior classification performance compared to dMHL in distinguishing pre- versus post-transplant cfDNA samples. MPCI achieved significantly higher specificity (MPCI: 0.990$\pm$0.010 vs. dMHL: 0.893$\pm$0.042, p=0.031) and accuracy (MPCI: 0.868$\pm$0.023 vs. dMHL: 0.768$\pm$0.027, p=0.014), with a non-significant trend toward higher sensitivity (MPCI: 0.791$\pm$0.038 vs. dMHL: 0.692$\pm$0.050, p=0.139) and higher AUC (MPCI: 0.961$\pm$0.013 vs. dMHL: 0.934$\pm$0.015, p=0.328) (Fig 3c). ROC curve was generated by aggregating probability predictions across all folds (Fig 3d). Results suggest MPCI's added benefits in real-world liquid biopsy applications.

## Investigating DMR prioritization for biomarker ranking and selection

Next, we set out to evaluate MPCI's utility in biomarker selection. Toward this end, we ranked biomarkers in multiple contexts according to various metrics and compared the outputs. A set of previously identified Differentially Methylated Regions (DMRs) were used for this task. These DMRs were derived from two sources: (1) DMRs specific to neuron cell types identified by prior epigenetic studies [27], and (2) DMRs we identified through the analysis of methylation array

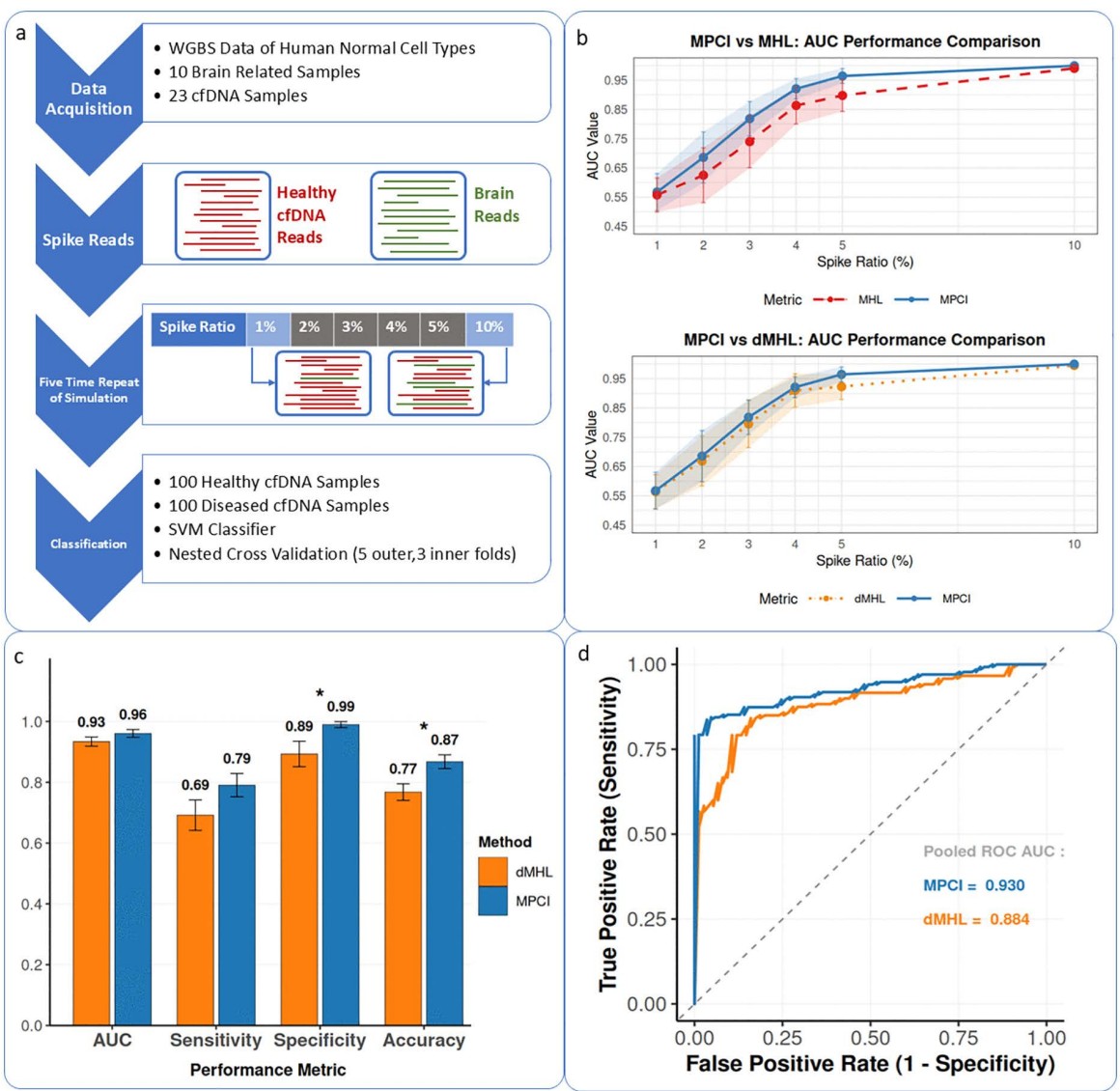

**Fig 3. a) Workflow of metric evaluation in simulated cfDNA data.** Public methylation sequencing data from healthy human cell types was used, with brain tissue and cfDNA samples selected to simulate disease conditions. In silico spike-in ratios (1%, 2%, 3%, 4%, 5%, 10%) were applied by randomly substituting reads from brain tissue into healthy cfDNA reads. For each ratio, 100 disease condition samples were generated, while 100 healthy condition samples were created by random sampling from healthy cfDNA. Metrics (MPCI, MHL and dMHL) were calculated for selected random regions, and an SVM classifier was trained on the data using nested cross validation. Performance metrics (AUC, accuracy, specificity, sensitivity) were evaluated. This process from sample simulation to classification was repeated five times. b) Performance comparison of MPCI, MHL and dMHL across simulated spike-in ratios. Line plots showing AUC values for MPCI (blue), dMHL (orange), and MHL (red) across varying spike-in concentrations. Error bars and shaded regions represent standard deviation from five experimental replicates. c) Comparison of classification performance metrics between dMHL (orange) and MPCI (blue) in distinguishing pre- versus post-transplant cfDNA methylation profiles. Values represent mean performance across 5-fold nested cross-validation. Error bars indicate standard error across cross-validation folds. Asterisks denote statistical significance (*p < 0.05) based on Wilcoxon rank-sum tests with Benjamini-Hochberg correction for multiple comparisons. d) ROC curves comparing classification performance of dMHL (orange) and MPCI (blue) in pre- versus post-transplant cfDNA methylation profiles aggregated across 5-fold nested cross-validation. The pooled AUC values represent the area under the aggregated ROC curves across folds, which differs from the mean AUC values in panel c because it is calculated from pooled predictions rather than averaging fold-wise AUCs.

datasets comparing brain and blood tissues. We selected these regions in 95 whole blood and 567 brain samples [30] and calculated MPCI and MHL for all regions. To establish a scoring scheme, the metric values for each DMR were averaged across samples within each cell type, and the difference between the averages for Neuron and cfDNA was computed. This approach ensures that regions with consistent differential methylation patterns are prioritized (Fig 4a). The MHL differential (range 0–1) was multiplied by two to align its scale with that of the MPCI differential (range 0–2) for direct visual comparison. Plotting these differential scores for all DMRs revealed an overall linear agreement between the two metrics (Fig 4b). To investigate discordant prioritization, we segmented the scatter plot into four quadrants: **Region 1**, containing markers with high differential MPCI but low MHL scores (n = 51); **Region 2**, with high differential score markers in both metrics (n = 36); **Region 3**, with markers low in both (n = 939); and **Region 4**, with markers exhibiting low differential MPCI but high MHL scores (n = 6). We focused our analysis on Regions 1 and 4, which contained DMRs prioritized exclusively by one metric. Within these regions, we selected the most divergent outlier markers—the points farthest from the diagonal line—for detailed investigation.

For each selected outlier, we identified a comparative marker by selecting the farthest point in the scatter plot that shared a similar MPCI value but had a very different MHL value (or vice versa). This paired-comparison allowed us to directly investigate why certain regions are strongly prioritized by one metric but not the other. In Region 1, the most divergent outlier, Marker I (high MPCI score/low MHL score) was compared with a marker with a similar high MPCI but a concordantly high MHL (Marker II). Despite nearly identical MPCI differential scores (~1.45), their MHL differential scores were very different (0.41 vs. 1.56; Fig 4c). Visualization of methylation patterns in read data showed both as strong candidates, but Marker I's MHL score was penalized due to sparse unmethylated signals that disrupted the fully methylated haplotypes measured by MHL's formulation. This would lead to a false exclusion of a good marker. Conversely, in Region 4, we excluded the most extreme outlier (MarkerX – chr1:143916766–143917144 (hg19)) due to extremely low cfDNA read coverage (Average coverage in 23 cfDNA samples = 0.48X). We therefore analyzed the next most divergent outlier (Marker III, low MPCI score/high MHL score), comparing it to a marker with a similar high MHL but a concordantly high MPCI (Marker IV). These two markers had near-identical MHL differential scores (~1.26) but divergent MPCI scores (0.81 vs. 1.63; Fig 4d). Inspection revealed Marker III's high MHL differential score was indeed an artifact: partial methylation haplotypes in neurons produced a low neuronal MHL, while the fully methylated reads in cfDNA inflated its cfDNA MHL. In both cases, MPCI's consistency based formulation provided a more accurate and robust quantification of differential methylation, effectively reducing the false negatives and false positives inherent to MHL's fully methylated haplotype dependent scoring.

### Investigating DMR prioritization for disease detection

To assess MPCI's performance in region prioritization for disease detection, we simulated 100 healthy and 100 diseased cfDNA samples using oligodendrocyte and cfDNA methylation data [27]. A 10% spike-in of oligodendrocyte reads was introduced into healthy cfDNA to mimic disease conditions. The DMRs used for this analysis included those specific to oligodendrocytes [27] and those associated with brain tissue, identified from methylation array datasets [30]. These regions served as input for region prioritization tests on the 200 simulated samples. MPCI and MHL were calculated for each region across all samples, averaged within each group, and the differences between the two groups were computed. The DMRs were then sorted based on these differences, and the top 10 regions were selected (Fig 5a).

We next analyzed markers where MPCI and MHL prioritized differently. Fig 5b shows a region ranked 7th by MPCI and 15th by MHL. In this region, the oligodendrocyte sample shows a strong shift towards fully methylated haplotypes (high MHL) while the cfDNA sample maintains a distinct population of fully unmethylated haplotypes (high uMHL). This balanced, opposing shift in haplotype extremes—which would also yield extreme dMHL values—is effectively captured by MPCI, validating its sensitivity to biologically relevant differential methylation. In contrast, Fig 5c depicts an example where MHL's preferentially ranked a region compared with MPCI (14th). Here, inspection of reads show that the signal is driven

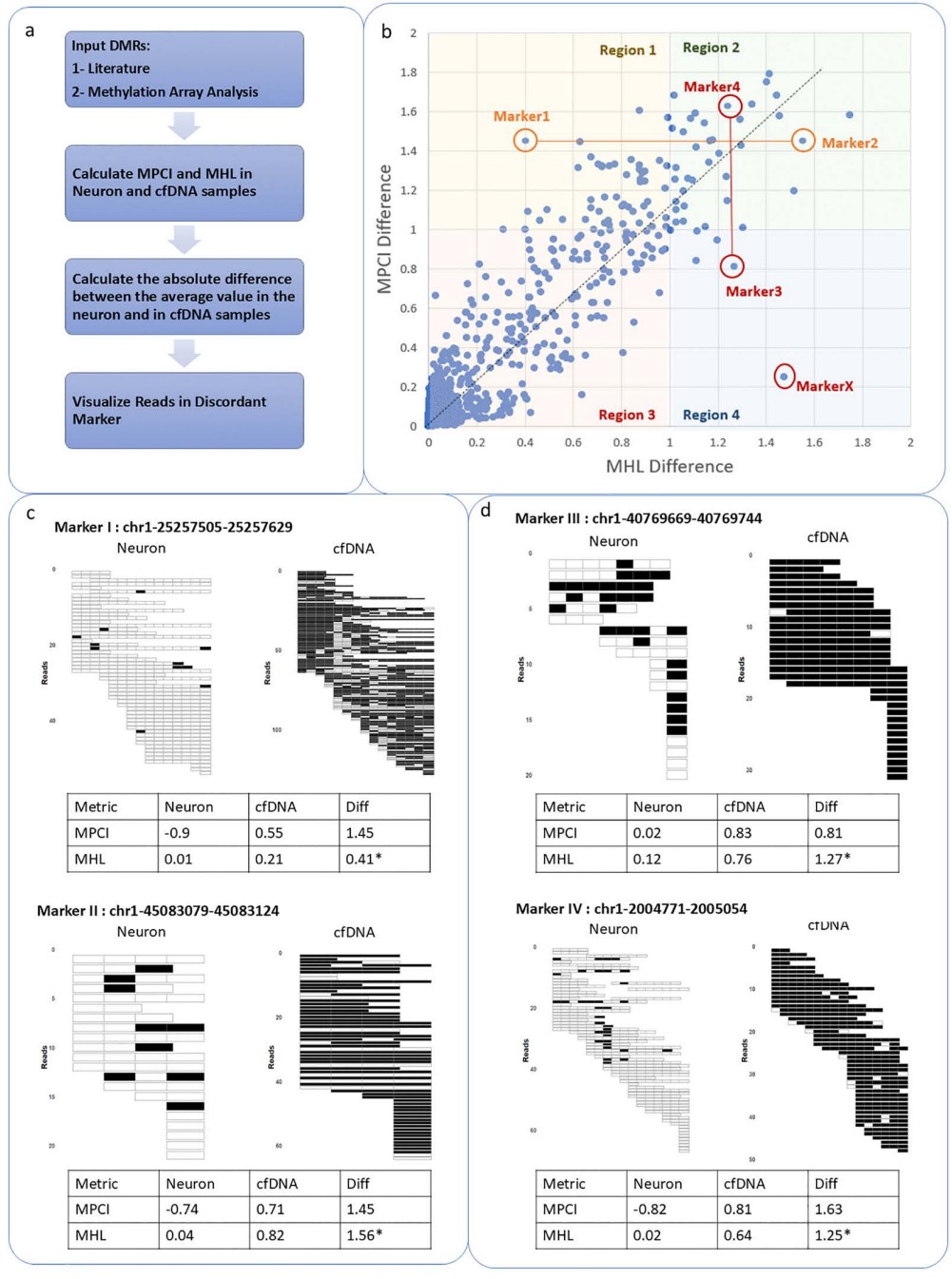

**Fig 4. a) Schematic workflow for identifying and prioritizing differentially methylated regions (DMRs) using MPCI vs. MHL. b) Scatter plot of differential methylation scores for input DMRs.** The plot is divided into quarters to investigate discordant markers. c) Example markers where MPCI assigns the same difference score to cfDNA and neuron samples, while MHL shows a stronger differentiation. Methylation patterns are displayed for one representative sample of oligodendrocyte and cfDNA, with black indicating methylated CpG sites and white indicating unmethylated CpG sites. The average MPCI and MHL values for each group and the difference between groups are represented. The marker position is stated based on hg19. The illustrated regions are example samples of neuron and cfDNA WGBS data. *MHL difference value is doubled to become comparable with MPCI scale. d) Example marker where MHL assigns the same difference score to cfDNA and neuron samples, while MPCI shows a stronger differentiation.

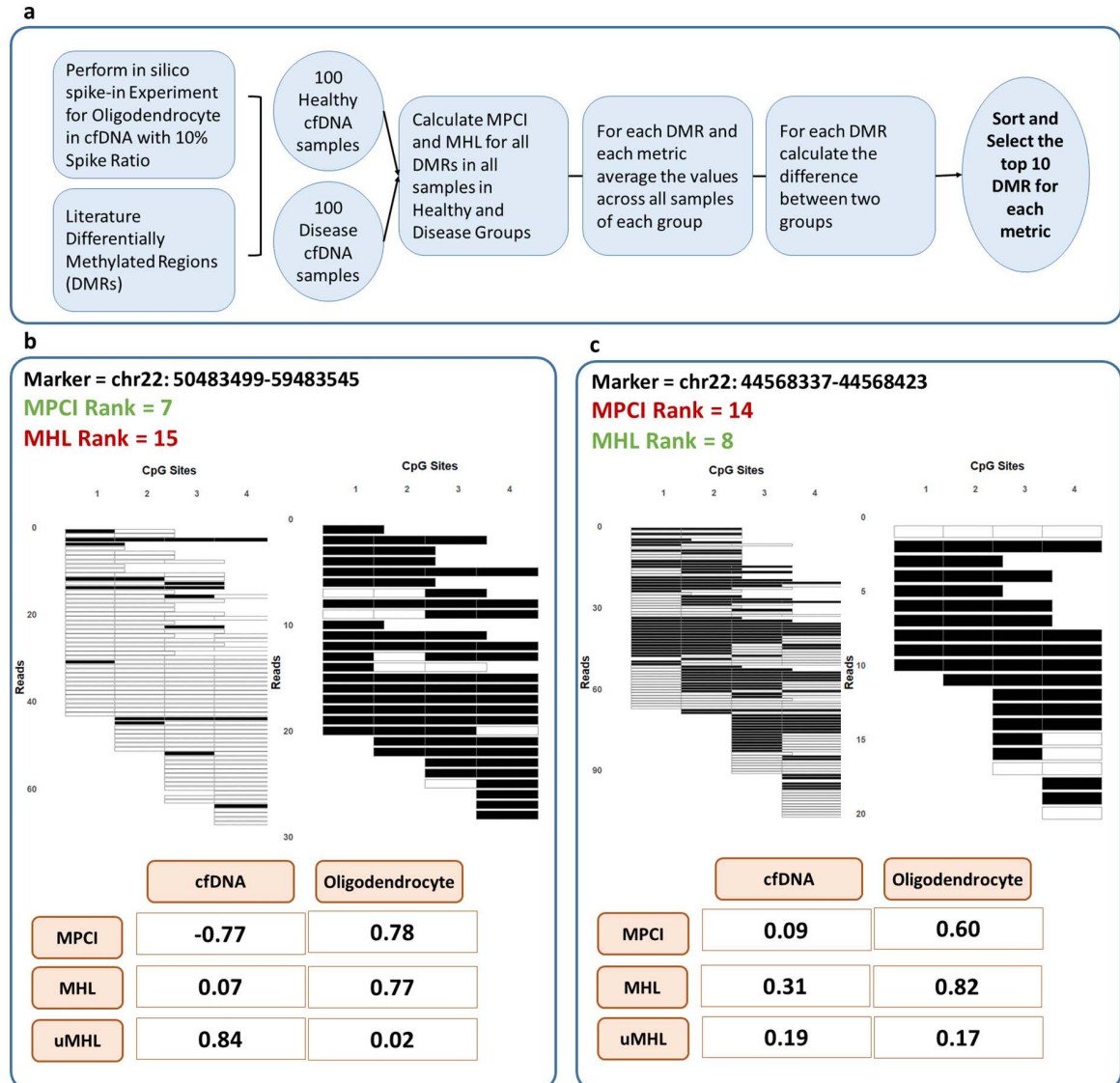

**Fig 5. a) Schematic workflow for identifying disease-associated DMRs.** The process involves a 10% spike-in of oligodendrocyte-derived bisulfite sequencing reads into cfDNA reads to mimic disease conditions, followed by MPCI and MHL calculation for each DMR. DMRs are ranked based on differences between groups, and the top 10 DMRs are selected separately for MPCI and MHL. b) Example DMR ranked in the top 10 by MPCI but not by MHL. Methylation patterns in the DMR for one cfDNA and one oligodendrocyte sample are shown, with black representing methylated CpG sites and white representing unmethylated CpG sites. c) Example DMR ranked in the top 10 by MHL but not by MPCI. Methylation patterns for one cfDNA and one oligodendrocyte sample are visualized similarly.

more unilaterally, lacking the same degree of contrast in both methylated and unmethylated haplotypes. This discrepancy demonstrates the limitations of using MHL (or uMHL) in isolation and underscores the need for a metric like MPCI that integrates information from both ends of the methylation spectrum for reliable region prioritization. Finally, to assess whether using dMHL is generally equivalent to MPCI for marker selection, we performed the same systematic comparison with dMHL. When we ranked all regions by dMHL and compared the resulting top 10 markers to the MPCI top 10, we

found an intersection of only 6 regions. This indicates that while the two metrics correlate and generally agree, they are not practically identical for prioritization.

## Discussion

In this study, we introduce the Methylation Pattern Consistency Index (MPCI), a novel and symmetric metric for quantifying DNA methylation patterns from read-level NGS data. Through extensive benchmarking against established metrics (including MHL, uMHL, and a panel of heterogeneity measures (ME, EP, PDR, FDRP, qFDRP)) we demonstrate that MPCI offers a more sensitive and robust quantification of methylation consistency. MPCI was specifically designed to address two fundamental limitations of the widely used MHL metric. First, MHL focuses exclusively on fully methylated haplotypes and requires a separate measurement (uMHL) to capture unmethylated patterns, making its application asymmetric and dependent on prior knowledge of methylation direction. Second, MHL quantifies co methylation within individual reads but does not explicitly assess consistency across reads at the same locus. By integrating weighted Manhattan similarities across both reads and CpG sites, MPCI provides a single, symmetric score that captures consistent patterns—whether methylated or unmethylated—while simultaneously accounting for read-to-read concordance. This design enables MPCI to detect subtle, consistent signals even when they are not perfectly consecutive across all CpGs, a feature particularly valuable for detecting rare tissue derived haplotypes in liquid biopsy.

Our comparative analyses confirm MPCI's superior performance across multiple contexts. In cell type classification, MPCI outperformed all seven alternative metrics in distinguishing closely related CD4 and CD8 T cells (AUC 0.915) and achieved higher multi tissue classification accuracy (92.25%). Even when evaluated in Methylation Haplotype Blocks (genomic regions pre-selected for high co methylation where MHL is expected to excel.), MPCI maintained a significant advantage in AUC and sensitivity, underscoring its robustness. When compared to the symmetric derivative differential MHL (dMHL = MHL − uMHL), MPCI showed significantly higher discriminative power, confirming that it captures information beyond a simple arithmetic combination of methylated and unmethylated haplotype loads.

In liquid biopsy simulations, MPCI consistently outperformed both MHL and dMHL across spike in ratios from 1% to 10%, demonstrating enhanced sensitivity at low disease derived DNA fractions and greater measurement stability. Most importantly, in a real-world clinical validation using pre and post liver transplant cfDNA samples, MPCI achieved significantly higher accuracy and specificity than dMHL, proving its practical utility in a biologically complex, low signal background.

Despite its advantages, MPCI's computational complexity scales quadratically with read depth and region size, which may limit its direct application to extremely high coverage whole genome datasets without appropriate optimization. However, for targeted biomarker regions this cost is manageable. Future work should validate MPCI in additional clinical cohorts, especially in early stage cancers and minimal residual disease monitoring, where tumor derived cfDNA fractions are often below 1%. Overall, we emphasize that diverse properties of region-wise DNA methylation states are deducible from read-level NGS data, and quantification in a way to best integrate them into a single numerical value is critical. The way DNA methylation is captured and quantified in genomic regions consisting of multiple CpGs has major effects on the output and results of downstream analyses, be it biomarker discovery, DMR prioritization, or sample classification for diagnosis. Hopefully this and similar studies can convince researchers to approach the data quantification step with more care in their future epigenetic studies.

## Methods

### Data collection, processing and metric calculation

We obtained a comprehensive dataset of Whole Genome Bisulfite Sequencing (WGBS) from GSE186458 comprising methylation profiles from various tissues [27]. In the original study, paired-end FASTQ files were mapped to the human (hg19 and hg38), lambda, pUC19 and viral genomes using bwa-meth (v.0.2.0) [31] then converted to BAM files using

SAMtools (v.1.9) [32]. Duplicated reads were marked by Sambamba (v.0.6.5) [33], and reads with low mapping quality, duplicates, or those not mapped in a proper pair were excluded. Reads were then stripped of non-CpG nucleotides and converted to PAT format using wgbstools (v.0.1.0) [34]. For our analysis, we directly downloaded these processed PAT files from GEO [27]. Multiple differential methylation analysis scenarios were considered using this dataset. For the evaluation of cell type classification contexts, blood T cell (CD4 and CD8), brain (oligodendrocyte, cortex, neuron and cerebellum), liver, lung, colon and bone marrow samples were selected from this dataset.

To simulate liquid biopsy scenarios, in-silico spike-in experiments were conducted using WGBS data from healthy cfDNA and neuron samples obtained from the same dataset [27]. Reads from target tissue were randomly selected, and a percentage of reads (1%, 2%, 3%, 4%, 5%, and 10%) were substituted with reads from the healthy cfDNA. This approach mimicked varying levels of disease-specific cfDNA in a background of healthy cfDNA, allowing for the evaluation of the proposed metric under conditions representative of real-world liquid biopsy applications.

Our evaluations used 400 randomly selected regions with diverse range of CpG numbers (100 regions: 3–7 CpGs, 100 regions: 8–12 CpGs, 100 regions: 13–17 CpGs, 100 regions: 18–22 CpGs). 8 regions with genomic overlap were filtered from the downstream analysis. These regions were selected to ensure a robust evaluation of the proposed metric.

To evaluate MPCI in real liquid biopsy data, cell-free DNA methylation data from liver transplant patients were obtained from GEO accession GSE262275 [29], comprising 130 blood samples from 44 patients at various timepoints. For this dataset, the original study's processing included trimming paired-end FASTQ files with TrimGalore (v.0.6.6) [35] and mapping them to the human genome (GRCh37/hg19) using Bismark (v.0.22.3) [36]. The resulting BAM files were processed with SAMtools (v.1.12) [32], and reads were subsequently converted to PAT format using wgbstools (v.0.1.0) [34]. We downloaded these pre-computed PAT files from GEO and focused on pre-transplant (POD0-PRE, n = 27) and post-operative day 0 (POD0-POST, n = 31) samples and performed our analysis on the 400 most covered regions in the dataset (see performance evaluation subsection).

All WGBS data were therefore downloaded from GEO in PAT file format [27,29]. The PAT file format stores read-level DNA methylation information and serves as a compact methylation-oriented representation of a BAM file. Each row in a PAT file contains the chromosome, starting CpG index, a methylation pattern string across consecutive CpG sites (with C for methylated, T for unmethylated, and '.' for unknown status), and the count of reads exhibiting that exact pattern.

Using these PAT files as input, we calculated the key methylation metrics used in our analysis. The methylation quantifying metrics including Methylation Haplotype Load (MHL), Methylation Entropy (ME), Epi-polymorphism (EP), Proportion of Discordant Reads (PDR), Fraction of Discordant Read Pairs (FDRP), and Quantitative FDRP (qFDRP) were computed following the mathematical formulations described by [19]. Unmethylation Haplotype Load (uMHL) was calculated based on same mathematical formulation of MHL adjusted for fully unmethylated haplotypes. MPCI, our introduced metric, was calculated using a custom R function that implements the formula (see MPCI metric definition subsection). R script for the calculation of all the metrics is deposited in https://github.com/NaghmeNazer/MPCI/tree/main/cell_type_classification_contexts.

## Differentially Methylated Region (DMR) identification

To compare the metrics for scoring and prioritizing previously selected biomarkers, we first identified differentially methylated regions (DMRs) between the cell types of interest. One set of selected DMRs was derived from whole-genome bisulfite sequencing (WGBS) data used in our analysis. This study introduced a set of regions for each cell type, and we selected the top 1000 DMRs specific to oligodendrocytes [27].

An independent set of DMRs was identified by analyzing a methylation array dataset of brain and blood samples obtained from GSE43414 [30]. This dataset contains methylation profiles for both tissue types. To ensure data quality, we first filtered out problematic probes, including: [1] probes with a detection p-value greater than 5%, [2] probes showing negative intensity values, [3] probes located within SNPs with an allele frequency above 5% (as these may indicate SNP

occurrence rather than site-specific methylation), and [4] non-specific probes that may map to multiple genomic locations [37]. Low-quality samples were also excluded from our analysis. Which means samples with low median signal intensities in both methylated (M) and unmethylated (U) channels were removed by applying a log2 transformation and filtering out those with median values below 10 in both signals.

After preprocessing the data, we employed ipDMR [38] to detect DMRs between brain and blood samples. These identified DMRs were then combined with the oligodendrocyte-specific regions and used for region prioritization experiments.

**MPCI metric definition**

The Methylation Pattern Consistency Index (MPCI) was developed to capture consistent methylation patterns across sequencing reads. Consecutive CpG sites within a sequencing read were modeled as a binary signal, with methylated CpGs represented as 1s and unmethylated CpGs as 0s. In cases where the methylation status of a CpG site was unknown—such as at the starts or ends of reads, or when explicitly labeled as unknown in the data—these sites were assigned NA (not available) values. This transformation resulted in a binary matrix R, where rows correspond to reads and columns correspond to consecutive CpG sites (Fig 1a).

To compute MPCI, we employed Manhattan similarity, defined as:

$$signal_i = row_i\ in\ matrix\ R \tag{1}$$

$$ManhattanDistance\left(Signal_i, Signal_j\right) = L1\ norm\ of\ Signal_i\ and\ Signal_j \tag{2}$$

$$S_{i,j} = ManhattanSimilarity\left(Signal_i, Signal_j\right) = 1 - \frac{ManhattanDistance(Signal_i, Signal_j)}{Number\ of\ Columns\ in\ Matrix\ R} \tag{3}$$

The similarity was calculated for all pairwise combinations of rows and columns in R. Weights were assigned based on the proportion of methylated (1s) and unmethylated (0s) CpGs in each pairwise comparison. If the number of 1s exceeded the number of 0s, a weight of +1 was assigned; if 0s exceeded 1s, a weight of -1 was assigned. In cases where the number of 1s and 0s were equal, the weight was randomly assigned as +1 or -1.

$$W_{i,j} = \begin{cases} +1 & th > 0.5 \\ -1 & th < 0.5 \\ random + 1\ or - 1 & th = 0.5 \end{cases} \tag{4}$$

$$th = \frac{Total\ number\ of\ 1s\ in\ Signal_1\ and\ Signal_2}{Total\ number\ of\ 1s\ and\ 0s\ in\ Signal_1\ and\ Signal_2} \tag{5}$$

The weighted average of the Manhattan similarities was computed row-wise and column-wise:

$$MS = \frac{\sum_{i=1}^{n} \sum_{j=1}^{n} W_{i,j} \times S_{i,j}}{\sum_{i=1}^{n} \sum_{j=1}^{n} W_{i,j}} \tag{6}$$

and the final MPCI metric was calculated as:

$$MPCI = \frac{MS(R) + MS(R^T)}{2} \tag{7}$$

NA values are systematically handled during calculations. When computing thresholds or means, NA values are ignored. If a calculation results in a NaN due to all values being NA, the output is set to NA. During similarity computations, NA values are excluded from distance metrics and summations. If the resulting data is entirely composed of NA values or empty, the output is appropriately set to NA. This approach ensures that missing data does not disrupt the analysis, maintaining the reliability of the calculations. The pseudocode for this calculation is available in supplementary material (S4 Fig). Due to pairwise comparisons between columns and rows of matrix R, MPCI scales quadratically with read depth (n) and region CpG number (m) ($O(n^2 + m^2)$).

To evaluate the impact of random weight assignment (±1 when th = 0.5), we performed 100 replicate MPCI calculations on 400 genomic regions (ranging from 3 to 22 CpGs) using real WGBS data from multiple tissues [27]. Random weight assignments occurred in 6.6% of row-wise and 1.1% of column-wise comparisons. Across replicates, MPCI values showed minimal variation (mean SD = 0.005), demonstrating that stochastic weight assignment has negligible effects on results.

MPCI values range from -1 to +1, where negative values indicate consistent unmethylated patterns, positive values indicate consistent methylated patterns, and values near zero indicate random methylation patterns.

**Performance evaluation**

**(1) Distinguishing methylation patterns between different cell types. Real Methylation Data (CD4 vs. CD8)**: The selected random regions were used for this test experiment. For each of these regions, eight methylation quantifying metrics (Methylation Haplotype Load (MHL), Unmethylation Haplotype Load (uMHL), Methylation Entropy (ME), Epipolymorphism (EP), Proportion of Discordant Reads (PDR), Fraction of Discordant Read Pairs (FDRP), and Quantitative FDRP (qFDRP)) were calculated for 10 CD4 and 10 CD8 samples. A Support Vector Machine (SVM) classifier with linear kernel was trained using 5 outer fold and 3 inner fold nested cross-validation on the data and the hyperparameter C was tuned in the training process. Performance metrics (AUC, accuracy, sensitivity, specificity) were evaluated. To check how sensitive our analysis is on selected random regions, the same classification analysis was set and performed on Methylation Haplotype Blocks [15]. In another experiment, the same approach was applied to a multiclass classification task involving five tissue types: brain, bone marrow, lung, liver, and colon. A multiclass SVM classifier with linear kernel and default hyper parameter (cost = 1) was trained using 5-fold cross-validation on 80% of the data consisting of MPCI and MHL values calculated on 323 random regions (regions with more than 10x coverage in all samples of this test scenario from the initial 400 random regions), and classification accuracy was measured on the remaining 20%. This process was repeated 100 times, and the accuracy values were averaged to compare the performance of MPCI and MHL in quantifying methylation. Then to compare the performance of MPCI with MHL formulation in a symmetric manner, we defined dMHL = MHL – uMHL. And further analysis was performed to compare MPCI with dMHL.

**(2) Detecting low-abundance disease-specific cfDNA in simulated and real liquid biopsy scenarios. Simulated Liquid Biopsy Scenarios**: Methylation data from healthy brain and cfDNA samples were used to conduct in-silico spike-in experiments at ratios of 1%, 2%, 3%, 4%, 5%, and 10%. For each ratio, healthy cfDNA samples were generated by randomly sampling reads to achieve a coverage of 100x. Subsequently, based on the specified spike-in ratio, a corresponding percentage of reads from brain tissue samples were randomly selected and substituted into the healthy cfDNA reads. This process created simulated datasets representing varying levels of disease-specific cfDNA in a background of healthy cfDNA. MPCI, dMHL and MHL were then calculated for randomly selected genomic regions to evaluate their performance in detecting low-abundance methylation signals. These regions were selected from the initial 400 random regions as to have more than 10X coverage in all samples. An SVM classifier with linear kernel and nested cv training approach was used (5 outer and 3 inner folds) for hyperparameter tuning (cost parameter C: $10^{-3}$ to 10) and evaluation of metrics' performance. Performance metrics (AUC, accuracy, sensitivity, specificity) were calculated and compared.

**Real Liquid Biopsy Samples**: Cell-free DNA methylation data from liver transplant patients were obtained from GEO accession GSE262275, comprising 130 blood samples from 44 patients at various timepoints. We focused on pre-transplant (PRE, n = 27) and post-operative day 0 (POST, n = 31) samples. High-coverage genomic regions were identified using a greedy algorithm requiring ≥50 reads per CpG across at least 5 consecutive CpGs. From these, the top 400 most covered regions on chromosome 22 were selected for downstream analysis. MPCI and dMHL were computed for each region. After removing features with >10% missing values and samples with >80% missing data (resulting in 44 samples/251 features for MPCI and 39 samples/361 features for dMHL), we implemented a nested cross-validation framework for classification. An SVM was trained using 5-fold outer cross-validation for performance estimation and 3-fold inner cross-validation for hyperparameter optimization (cost parameter C: $10^{-3}$ to $10^{3}$). Receiver operating characteristic (ROC) curves were generated by aggregating probability predictions across all folds, calculating true positive and false positive rates at varying classification thresholds. Performance metrics (AUC, sensitivity, specificity, accuracy) were compared between MPCI and dMHL using Wilcoxon rank-sum tests with paired observations across cross-validation folds. Statistical significance was determined at $p < 0.05$, with adjustment for multiple comparisons using the Benjamini-Hochberg method. Confidence intervals for mean AUC values were calculated using standard error estimates from cross-validation folds.

## Supporting information

**S1 Fig. PCA plot of CD4 and CD8 cell based on all methylation quantifying metrics investigated in this study.**
(TIF)

**S2 Fig. Comparison of MPCI and MHL in multi-class classification.**
(TIF)

**S3 Fig. Scatter plot of MPCI vs dMHL values for CD4 and CD8 cell types.**
(TIF)

**S4 Fig. Pseudo-code for calculating MPCI in a genomic region.**
(TIF)

**S1 Table. Comprehensive performance comparison of MPCI vs. other methylation metrics on CD4 CD8 classification task.**
(DOCX)

**S2 Table. Analysis on MHB regions on CD4 CD8 classification task.**
(DOCX)

**S3 Table. Performance comparison of MPCI, dMHL, and MHL across spike-in ratios.**
(DOCX)

## Acknowledgments

We thank the University of Tehran (UT) and Sharif University of Technology (SUT) for server and computing resources.

## Author contributions

**Conceptualization:** Naghme Nazer, Mahya Mehrmohamadi.

**Data curation:** Naghme Nazer.

**Formal analysis:** Naghme Nazer.

**Funding acquisition:** Hoda Mohammadzade, Mahya Mehrmohamadi.

**Investigation:** Naghme Nazer.

**Methodology:** Naghme Nazer, Hoda Mohammadzade, Mahya Mehrmohamadi.

**Project administration:** Mahya Mehrmohamadi.

**Software:** Naghme Nazer.

**Supervision:** Hoda Mohammadzade, Mahya Mehrmohamadi.

**Visualization:** Naghme Nazer, Mahya Mehrmohamadi.

**Writing – original draft:** Naghme Nazer, Mahya Mehrmohamadi.

**Writing – review & editing:** Naghme Nazer, Hoda Mohammadzade, Mahya Mehrmohamadi.

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
