## [Decision Letter · Decision Letter 0]

9 Nov 2025

PCOMPBIOL-D-25-00973

MPCI: A novel metric for quantifying DNA methylation patterns in NGS data

PLOS Computational Biology

Dear Dr. Mehrmohamadi,

Thank you for submitting your manuscript to PLOS Computational Biology. After careful consideration, we feel that it has merit but does not fully meet PLOS Computational Biology's publication criteria as it currently stands. Therefore, we invite you to submit a revised version of the manuscript that addresses the points raised during the review process.

We look forward to receiving your revised manuscript.

Kind regards,

Ilya Ioshikhes

Section Editor

PLOS Computational Biology

Ilya Ioshikhes

Section Editor

PLOS Computational Biology

**Journal Requirements:**

At this stage, the following Authors/Authors require contributions: Hoda Mohammadzade, Naghme Nazer, and Mahya Mehrmohamadi. Please ensure that the full contributions of each author are acknowledged in the "Add/Edit/Remove Authors" section of our submission form.

3) Some material included in your submission may be copyrighted. According to PLOSu2019s copyright policy, authors who use figures or other material (e.g., graphics, clipart, maps) from another author or copyright holder must demonstrate or obtain permission to publish this material under the Creative Commons Attribution 4.0 International (CC BY 4.0) License used by PLOS journals. Please closely review the details of PLOSu2019s copyright requirements here: PLOS Licenses and Copyright. If you need to request permissions from a copyright holder, you may use PLOS's Copyright Content Permission form.

Potential Copyright Issues:

i) Figures 1A, 2A, 3A, and 4A. Please confirm whether you drew the images / clip-art within the figure panels by hand. If you did not draw the images, please provide (a) a link to the source of the images or icons and their license / terms of use; or (b) written permission from the copyright holder to publish the images or icons under our CC BY 4.0 license. Alternatively, you may replace the images with open source alternatives. See these open source resources you may use to replace images / clip-art:

4) In the online submission form, you indicated that  cross-validation, and spike-in process can be shared upon request.. All PLOS journals now require all data underlying the findings described in their manuscript to be freely available to other researchers, either

1. In a public repository

2. Within the manuscript itself

3. Uploaded as supplementary information.

2) If any authors received a salary from any of your funders, please state which authors and which funders..

**Reviewers' comments:**

Reviewer's Responses to Questions

**Comments to the Authors:**

Reviewer #1: DNA methylation abnormality is vital to many diseases. Due to the nature of DNA methylation, which would only occur in CpG sites, focusing on methylation haplotype rather than individual sites can be more beneficial in consistently distinguishing between the healthy and abnormal methylation profiles. The author introduced a new metric MPCI which overcome the disadvantages of current metrics such as MHL and showed more superior performance in distinguishing different cell types as well as better disease detection ability from simulated cell-free DNA.

Major comments:

What is the rationale of summing up the MS(R) and MS(RT)? I understand that the author want to also consider the consistency among different reads. But since the distance for rows represent the similarity among different reads and the distance for columns represent the similarity among the CpG sites, it seems not distinguishing between the difference of the different CpG sites in the candidate region like what MHL does. Is there any biological evidence that can prove that small portions of reads with consecutive CpGs should be considered the same level of methylation the same as the randomly methylated ones? (as the Figure 1 showed)

The Figure 4 is a little cherry-picked to me. Is there any example where the MHL can better distinguish than MPCI for some DMRs?

Also since the definition of MHL and MPCI is different and the range is also different, it seems to be not intuitive and convincing to directly compare the absolute difference of the exact values. MHL is sensitive to the length of the considered region and could be underestimated if there are more NAs and length is large.

When selecting the CpG with different numbers, do they overlap with each other? For a specific region, how to determine what CpG number is the best?

The author described some other metrics than the MHL, have you compared these metrics with MPCI?

Reviewer #2: Nazer et al. developed a simple yet effective metric, MPCI, to quantify DNA methylation patterns in regions where nearby CpGs are concordantly methylated or unmethylated. The authors demonstrated that MPCI outperforms existing metrics, particularly MHL, which only quantifies the level of co-methylation. The manuscript presents interesting findings, but the following issues should be addressed before it can be considered for publication in PLoS Computational Biology:

(1) The schematic plots in Figures 1B and 1C appear to be inconsistent. If M represents mean methylation in Figure 1B, it should equal 1 for the control region. Similarly, beta in Figure 1C should equal 0 for the control region. Please verify and correct these illustrations.

(2) MPCI appears to be mathematically equivalent to (MHL - uMHL). If we define dMHL = MHL - uMHL, how does its performance compare to MPCI? I recommend including dMHL in all comparative figures and providing a scatter plot showing the relationship between MPCI and dMHL.

(3) The authors consistently use 400 randomly selected regions for benchmarking. However, it is unclear how sensitive the results are to region selection. Given that MHL has been shown to perform optimally in MHB regions, could you repeat the analysis in Figure 2 using all MHB regions and compare the performance of MHL versus MPCI? Additionally, statistical significance tests are missing in Figure 2C and should be added.

(4) In Figure 3, since the samples are simulated, the authors should generate multiple independent cohorts (each with 100 healthy and 100 disease samples) rather than repeatedly resampling from a single cohort. This would provide a more robust assessment of performance variability.

(5) In Figure 5B, it appears that dMHL could achieve a similarly high ranking as MPCI. This observation warrants further discussion.

(6) To properly benchmark cfDNA-based cancer detection, the authors should validate their approach using real cfDNA methylation datasets rather than relying solely on simulated data. Several public datasets are available for this purpose, and their inclusion would significantly strengthen the manuscript's conclusions.

**Have the authors made all data and (if applicable) computational code underlying the findings in their manuscript fully available?**

The PLOS Data policy requires authors to make all data and code underlying the findings described in their manuscript fully available without restriction, with rare exception (please refer to the Data Availability Statement in the manuscript PDF file). The data and code should be provided as part of the manuscript or its supporting information, or deposited to a public repository. For example, in addition to summary statistics, the data points behind means, medians and variance measures should be available. If there are restrictions on publicly sharing data or code —e.g. participant privacy or use of data from a third party—those must be specified.requires authors to make all data and code underlying the findings described in their manuscript fully available without restriction, with rare exception (please refer to the Data Availability Statement in the manuscript PDF file). The data and code should be provided as part of the manuscript or its supporting information, or deposited to a public repository. For example, in addition to summary statistics, the data points behind means, medians and variance measures should be available. If there are restrictions on publicly sharing data or code —e.g. participant privacy or use of data from a third party—those must be specified.

Reviewer #1: Yes

Reviewer #2: **No:** No new data was generated in this study but the authors should make their scripts available.No new data was generated in this study but the authors should make their scripts available.

PLOS authors have the option to publish the peer review history of their article (what does this mean? ). If published, this will include your full peer review and any attached files.). If published, this will include your full peer review and any attached files.

**Do you want your identity to be public for this peer review?** For information about this choice, including consent withdrawal, please see our For information about this choice, including consent withdrawal, please see our Privacy Policy ..

Reviewer #1: No

Reviewer #2: **Yes:** Jiantao ShiJiantao Shi

**Figure resubmission:**
---

## [Decision Letter · Decision Letter 1]

20 Feb 2026

PCOMPBIOL-D-25-00973R1

MPCI: A novel metric for quantifying DNA methylation patterns in NGS data

PLOS Computational Biology

Dear Dr. Mehrmohamadi,

Thank you for submitting your manuscript to PLOS Computational Biology. After careful consideration, we feel that it has merit but does not fully meet PLOS Computational Biology's publication criteria as it currently stands. Therefore, we invite you to submit a revised version of the manuscript that addresses the points raised during the review process.

We look forward to receiving your revised manuscript.

Kind regards,

Ilya Ioshikhes

Section Editor

PLOS Computational Biology

Ilya Ioshikhes

Section Editor

PLOS Computational Biology

**Journal Requirements:**

**Reviewers' comments:**

Reviewer's Responses to Questions

**Comments to the Authors:**

Reviewer #1: The authors addressed the previously raised issues.

Reviewer #2: My comments were largely addressed. However, I found that the authors did not describe the tools used in the analysis or the parameters employed. For example, it is unclear how different read-level metrics were calculated; the authors should describe the tools for alignment, read filtering, and coverage thresholds in detail in the Methods section. In addition, the authors should make all their scripts publicly available to ensure reproducible research. Currently, I only see the R function for MPCI calculation.

**Have the authors made all data and (if applicable) computational code underlying the findings in their manuscript fully available?**

The PLOS Data policy requires authors to make all data and code underlying the findings described in their manuscript fully available without restriction, with rare exception (please refer to the Data Availability Statement in the manuscript PDF file). The data and code should be provided as part of the manuscript or its supporting information, or deposited to a public repository. For example, in addition to summary statistics, the data points behind means, medians and variance measures should be available. If there are restrictions on publicly sharing data or code —e.g. participant privacy or use of data from a third party—those must be specified.requires authors to make all data and code underlying the findings described in their manuscript fully available without restriction, with rare exception (please refer to the Data Availability Statement in the manuscript PDF file). The data and code should be provided as part of the manuscript or its supporting information, or deposited to a public repository. For example, in addition to summary statistics, the data points behind means, medians and variance measures should be available. If there are restrictions on publicly sharing data or code —e.g. participant privacy or use of data from a third party—those must be specified.

Reviewer #1: None

Reviewer #2: **No:** The authors should make all their scripts publicly available to ensure reproducible research.The authors should make all their scripts publicly available to ensure reproducible research.

PLOS authors have the option to publish the peer review history of their article (what does this mean? ). If published, this will include your full peer review and any attached files.). If published, this will include your full peer review and any attached files.

**Do you want your identity to be public for this peer review?** For information about this choice, including consent withdrawal, please see our For information about this choice, including consent withdrawal, please see our Privacy Policy ..

Reviewer #1: No

Reviewer #2: No

**Figure resubmission:**
---

## [Decision Letter · Decision Letter 2]

27 Feb 2026

Dear Dr. Mehrmohamadi,

We are pleased to inform you that your manuscript 'MPCI: A novel metric for quantifying DNA methylation patterns in NGS data' has been provisionally accepted for publication in PLOS Computational Biology.

Best regards,

Ilya Ioshikhes

Section Editor

PLOS Computational Biology

Ilya Ioshikhes

Section Editor

PLOS Computational Biology

Reviewer's Responses to Questions

**Comments to the Authors: 
Please note here if the review is uploaded as an attachment.**

Reviewer #2: I have no further comments.

**Have the authors made all data and (if applicable) computational code underlying the findings in their manuscript fully available?**

The PLOS Data policy requires authors to make all data and code underlying the findings described in their manuscript fully available without restriction, with rare exception (please refer to the Data Availability Statement in the manuscript PDF file). The data and code should be provided as part of the manuscript or its supporting information, or deposited to a public repository. For example, in addition to summary statistics, the data points behind means, medians and variance measures should be available. If there are restrictions on publicly sharing data or code —e.g. participant privacy or use of data from a third party—those must be specified.requires authors to make all data and code underlying the findings described in their manuscript fully available without restriction, with rare exception (please refer to the Data Availability Statement in the manuscript PDF file). The data and code should be provided as part of the manuscript or its supporting information, or deposited to a public repository. For example, in addition to summary statistics, the data points behind means, medians and variance measures should be available. If there are restrictions on publicly sharing data or code —e.g. participant privacy or use of data from a third party—those must be specified.

Reviewer #2: Yes

PLOS authors have the option to publish the peer review history of their article (what does this mean? ). If published, this will include your full peer review and any attached files.). If published, this will include your full peer review and any attached files.

**Do you want your identity to be public for this peer review?** For information about this choice, including consent withdrawal, please see our For information about this choice, including consent withdrawal, please see our Privacy Policy ..

Reviewer #2: No

---

## [Editor Report · Acceptance letter]

PCOMPBIOL-D-25-00973R2

MPCI: A novel metric for quantifying DNA methylation patterns in NGS data

Dear Dr Mehrmohamadi,

I am pleased to inform you that your manuscript has been formally accepted for publication in PLOS Computational Biology. Your manuscript is now with our production department and you will be notified of the publication date in due course.

With kind regards,

Anita Estes
